# The Relationship between the Mediterranean Diet and Vascular Stiffness, Metabolic Syndrome, and Its Components in People over 65 Years of Age

**DOI:** 10.3390/nu16203464

**Published:** 2024-10-12

**Authors:** Leticia Gómez-Sánchez, Marta Gómez-Sánchez, Luis García-Ortiz, Cristina Agudo-Conde, Cristina Lugones-Sánchez, Susana Gonzalez-Sánchez, Emiliano Rodríguez-Sánchez, Manuel A. Gómez-Marcos

**Affiliations:** 1Unidad de Investigación en Atención Primaria de Salamanca (APISAL), Centro de Salud de San Juan, Avenida Portugal 83, 2º Planta, 37005 Salamanca, Spain; leticiagmzsnchz@gmail.com (L.G.-S.); lgarciao@usal.es (L.G.-O.); cagudoconde@usal.es (C.A.-C.); cristinals@usal.es (C.L.-S.); gongar04@gmail.com (S.G.-S.); 2Servicio de Urgencias, Hospital Universitario de La Paz, Paseo de la Castellana, 261, 28046 Madrid, Spain; martagmzsnchz@gmail.com; 3Servicio de Hospitalización a Domicilio, Hospital Universitario Marqués de Valdecilla, Avenida de Valdecilla, s/n, 39008 Santander, Spain; 4Instituto de Investigación Biomédica de Salamanca (IBSAL), Hospital Universitario de Salamanca, Paseo de San Vicente, 182, 37007 Salamanca, Spain; 5Red de Investigación en Cronicidad, Atención Primaria y Promoción de la Salud (RICAPPS), Avenida Portugal 83, 2º P, 37005 Salamanca, Spain; 6Gerencia de Atención Primaria de Salud de Castilla y León (SACyL), Avenida Portugal 83, 2º Planta, 37005 Salamanca, Spain; 7Departamento de Ciencias Biomédicas y del Diagnóstico, Universidad de Salamanca, C. Alfonso X el Sabio, s/n, 37007 Salamanca, Spain; 8Facultad de Enfermería y Fisioterapia, Universidad de Salamanca, Campus Miguel de Unamuno, C. Donantes de Sangre, s/n, 37007 Salamanca, Spain; 9Departmento de Medicina, Universidad de Salamanca, C. Alfonso X el Sabio, s/n, 37007 Salamanca, Spain

**Keywords:** Mediterranean diet, metabolic syndrome, brachial-ankle pulse wave velocity, cardio-ankle vascular index, vascular stiffness

## Abstract

Objectives: The aim of the study was to examine the relationship between the Mediterranean diet (MD) and vascular stiffness and metabolic syndrome (MetS), as well as its components in individuals over the age of 65, overall and by sex. Methods: The subjects of the study were people over 65 years of age, with a full record of all variables analyzed from the EVA, MARK, and EVIDENT studies. Data from 1280 subjects with a mean age of 69.52 ± 3.58 years (57.5% men) were analyzed. The MD was recorded with the validated 14 item MEDAS questionnaire. MetS was defined following the guidelines of the joint scientific statement from the Programa Nacional de Educación sobre el Colesterol III. Vascular stiffness was evaluated with the VaSera VS-1500^®^ device by measuring the cardio-ankle vascular index (CAVI) and the brachial-ankle pulse wave velocity (baPWV). Results: The mean MEDAS score was 6.00 ± 1.90, (5.92 ± 1.92 in males, 6.11 ± 1.88 in females; *p* = 0.036). CAVI: 9.30 ± 1.11 (9.49 ± 1.05 males, 9.03 ± 1.13 females; *p* = <0.001). baPWV: 15.82 ± 2.56 (15.75 ± 2.46 males, 15.92 ± 2.68 females; *p* = <0.001). MetS was found in 51% (49% males, 54% females; *p* = 0.036). Subjects with MetS had lower MD adherence and higher vascular stiffness values than subjects without MetS. Overall, we found a negative association with MD score and the number of MetS components (β = −0.168), with glycemia (β = −0.007), triglycerides (β = −0.003), waist circumference (β = −0.018), CAVI (β = −0.196) and baPWV (β = −0.065), and a positive association with HDL cholesterol (β = 0.013). Regarding sex, associations followed the same direction but without reaching statistical significance with blood glucose and triglycerides in females and with HDL cholesterol and waist circumference in males. Conclusions: The results indicate that greater adherence to the Mediterranean diet decreases vascular stiffness and the percentage of subjects with MetS, although results differed in the association with MetS components by sex.

## 1. Introduction

Nutrition contributes to human health up to the final stages of life [1,2]. The Mediterranean diet (MD) has important health benefits, especially in people aged over 65 [3,4,5,6]. The MD is used in Mediterranean countries [7,8] and is generally rich in fresh foods such as fruits, vegetables, legumes, whole grains and nuts, all rich in fiber and antioxidants that improve cellular aging [9,10,11]. In people over 65, consuming MD components as part of a regular dietary pattern contributes to healthy aging [12,13,14] and plays a positive role in muscle function [15], in the prevention of depression [16], cardiovascular diseases (CVD) [17,18,19], type 2 diabetes [20], hypertension [21], obesity [22], in improving the lipid profile [23], in preventing cognitive decline and Alzheimer’s disease [24] and certain types of cancer [25]. It reduces morbidity, mortality [26] and fragility [27], and improves the quality of life [5]. For all these reasons, the MD was proclaimed a UNESCO Intangible Cultural Heritage in 2010 as a way of life that can increase hope and quality of life [28].

Vascular stiffness (VS) alters arterial elasticity, reducing the vascular wall’s ability to stretch and contract [29,30]. SV has the ability to predict cardiovascular diseases (CVD), and is similar to, or stronger than, other traditional cardiovascular risk factors [31], such as non-invasive measurements, with both brachial-ankle pulse wave velocity (baPWV) [32,33] and cardio-ankle vascular index (CAVI) [34,35]. VS partners with CVD, and is determined by age, gender, and blood pressure [29,31]; the relationship with the MD is unclear. Some studies have suggested that the MD probably does not behave as an independent risk factor for VS [36]. However, other studies have found that the MD improves arterial elasticity, thereby improving cardiovascular health [21].

Metabolic syndrome (MetS) is a combination of risk factors for arterial atherosclerosis, such as abdominal obesity, high blood pressure, high fasting plasma glucose (FPG) and atherogenic dyslipidemia [37]. MetS doubles CVD morbidity and mortality and carries a 1.5 times higher risk of mortality [38,39,40]. Greater fast food consumption, less time spent exercising and increased sedentary behavior during leisure time have led to the growing incidence of MetS [41]. In people over 65, MetS is more prevalent in females, with figures ranging from 35% to 50%; for males, these figures lie between 25% and 40%, and prevalence differs depending on the criteria used to define MetS. This is reflected in several studies [42], among them, one carried out in Singapore with 722 adults aged 65 years or over, which found a 41% prevalence of MetS as well as a 50% prevalence in those aged 85 years or older; and a higher prevalence in females than males [43]. In the ENRICA study [44], it was found that the MD, together with other Mediterranean lifestyles, was associated with a lower prevalence of MetS and a lower mortality due to the anti-inflammatory effect of the dietary components of the MD [45]. However, the effect of the MD on VS, MetS and each of its components has not been fully established in Caucasians aged over 65. With this study, we will provide new information on the relationship between the MD and arterial stiffness, assessed with two measures, CAVI and baPWV, in older Caucasian people. It is one of the first studies to analyze these aspects globally and by sex. The objectives of this work were, therefore: to study the relationship of the MD with VS, assessed with CAVI and baPWV, and with MetS and its components in Caucasians over 65 years of age, overall and by gender.

## 2. Methods

### 2.1. Type of Study

This is a cross-sectional descriptive study using variables collected in the following studies: EVA [46] (NCT02623894, registered 8 September 2016), MARK [47] (NCT01428934, registered 2 September 2011) and EVIDENT [48] (NCT02016014, registered 13 December 2013).

### 2.2. Population

The three projects were performed in primary care. For the present study, data were used from 1280 subjects aged over 65, with a full record of all the variables analyzed. The EVA study, in which selection was via random sampling of the urban population of 5 health centers without prior CVD at the start of the study (Reference people 43,946), contributed 135 of its 501 subjects [46]. From the MARK study, where the selection was made via random sampling among patients attending primary care consultations in 7 urban health centers and presenting intermediate cardiovascular risk, 948 of its 2511 subjects were incorporated. The EVIDENT study provided 197 of its 1104 subjects. In that study, the selection was made following a random sampling among patients attending a primary care center. Figure 1 shows the selection of subjects by origin.

A more extensive and detailed you are in previous publications of the three studies [46,47,48].

### 2.3. Ethics Committee and Participant Consent

The works were approved by the Drug Research Ethics Committee of Salamanca on the following dates: the EVA study [46], at the meeting of 4 May 2015, the MARK study [47], at the meeting of 3 December 2013, and the EVIDENT study [48], at the meeting of 25 April 2016. All persons included signed written consent. The recommendations set out in the Declaration of Helsinki [49] were followed throughout the project.

### 2.4. Variables and Measurement Methods

#### 2.4.1. Mediterranean Diet

MD adherence was valued using the 14 item MEDAS screener, validated in the Spanish adult population [50]. This questionnaire includes 12 questions that envelope the frequency of food consumption and 2 items that envelope the habits. The questions were punctuated with zero or one. A point was given for daily consumption of (a) ≥4 tablespoons of olive oil, (b) ≥2 servings of vegetables, (c) ≥3 pieces of fruit, (d) <1 serving of red or processed meat, (e) <1 serving of animal fat, (f) <1 cup of sugary carbonated drinks, (g) eating white meat in greater proportion to red meat. A point was also scored for weekly consumption of (a) ≥7 glasses of wine, (b) ≥3 servings of legumes, (c) ≥3 servings of fish, (d) ≥3 servings of nuts or dried fruit, (e) ≥2 servings of sofrito (a home-made sauce of onions and/or garlic and tomatoes, sautéed in extra-virgin olive oil), and (f) <2 baked goods. The final score range ranged from 0 to 14 points. We considered MD adherence for scores above 6 points, the median [50].

#### 2.4.2. Metabolic Syndrome

We considered that a patient had MetS when they met 3 or more criteria specified in Table 1 [37].

#### 2.4.3. Vascular Stiffness

The CAVI and baPWV stiffness parameters were measured with the VaSera VS-1500 device (Fukuda Denshi Co, Ltd., Tokyo, Japan), following the manufacturer’s instructions. Measurements were made with the patient in a supine position, after 5 min of rest, without drinking alcohol or smoking in the previous 10 min, in a quiet room, and without speaking during the duration of the examination. The electrodes were placed on both arms and both legs. A cardiac microphone was placed on the sternum and secured with adhesive tape. The device estimated the CAVI value automatically by substituting the stiffness parameters using the following equation: stiffness parameter β = 2 ρ × 1/(Ps − Pd) × ln (Ps/Pd) × PWV2, where ρ is blood density, Ps and Pd are SBP and DBP in mmHg, and PWV is measured between the aortic valve and the ankle. CAVI measurements were considered valid when at least 3 consecutive heartbeats were obtained [51]. The baPWV was calculated with the formula: baPWV = ((0.5934 × height (cm) + 14.4724))/tba, where tba is the time interval between arm and ankle waves.

#### 2.4.4. Cardiovascular Risk Factors

Weight was measured twice with electronic scales (Seca 770; Medical Scales and Measurement Systems, Birmingham, UK), with the participant lightly dressed and barefoot. Height was measured with a portable stadiometer (Seca 222; Medical Scales and Measurement Systems, Birmingham, UK), with the participant standing barefoot and the average of two readings being recorded. The body mass index (BMI) was estimated as weight (kg) divided by height (m) squared. WC was measured following the recommendations of the Spanish Society for the Study of Obesity [52].

Three blood pressure measurements were carried out. The average of the last two was taken as a reference measure. Measurements use an OMRON model M10-IT sphygmomanometer (Omron Health Care, Kyoto, Japan) [53]. Participants were considered to have hypertension if taking antihypertensive medication or had blood pressure levels ≥ 140/90 mmHg, to be diabetic if taking hypoglycemic drugs or having fasting plasma glucose levels ≥ 126 mg/dL or HbA1c levels ≥ 6.5%, and to have dyslipidemia if taking lipid-lowering drugs or having fasting total cholesterol levels ≥ 240 mg/dL, low-density lipoprotein cholesterol (LDL-c) ≥ 160 mg/dL, high-density lipoprotein cholesterol (HDL-c) < 40 mg/dL in males and <50 mg/dL in females, or triglycerides ≥ 150 mg/dL. Subjects were considered obese if they had BMI levels ≥ 30 kg/m^2^ [54].

### 2.5. Statistical Analysis

Continuous variables are shown as mean ± standard deviation. Categorical variables are shown as numbers and percentages. Differences between sexes or between subjects with and without MetS were calculated using the chi-square test for percentages and the Student’s t test for continuous variables.

To calculate the association between the mean MD score and the number of MetS components (as well as with each component individually) and VS, eight multiple linear regression models were performed with the MD score as an independent variable and the number of MetS components, SBP and DBP in mmHg, FPG in mg/dL, triglycerides in mg/dL, HDL cholesterol in mg/dL, WC in cm, CAVI and baPWV in m/s as dependent variables.

To estimate the association between MD adherence and the presence of MetS, its components, CAVI and baPWV, eight logistic regression models were performed. Adherence to the MD was the independent variable (encoded as MD adherence = 1, non-adherence = 0). Dependent variables were MetS (yes = 1, no = 0), BP ≥ 130/85 mmHg (yes = 1, no = 0), FPG ≥ 100 mg/dL (yes = 1, no = 0), TGC ≥ 150 mg/dL (yes = 1, no = 0), HDL-C mg/dL < 40 males, <50 females (yes = 1, no = 0) and WC ≥ 88 cm females, ≥102 cm males (yes = 1, no = 0). In all models, age, sex, and consumption of antihypertensive drugs (yes = 1, no = 0), hypoglycemic drugs (yes = 1, no = 0) and lipid-lowering drugs (yes = 1, no = 0) were included as adjustment variables. Because it is among the variables that have the greatest influence on vascular stiffness.

Analyses were performed overall, by sex, and in subjects with and without MetS. The program used to perform the analysis was SPSS Statistics for Windows, version 28.0 (IBM Corp., Armonk, NY, USA). We used a *p* value < 0.05 as the limit for statistical significance.

## 3. Results

### 3.1. Characteristics

Table 2 shows the variables analyzed overall and differences by gender. The mean MEDAS questionnaire score was 6.00 ± 1.90 (5.92 ± 1.92 males, 6.11 ± 1.88 females; *p* = 0.036). The mean CAVI value was 9.30 ± 1.11 (9.49 ± 1.05 males, 9.03 ± 1.13 females; *p* < 0.001). The mean baPWV was 15.82 ± 2.56 (15.75 ± 2.46 males, 15.92 ± 2.68 females; *p* < 0.001). MetS was found in 51% (49% males, 54% females; *p* = 0.036). Males had a higher percentage of the following MetS components: blood pressure ≥ 130/85 mmHg (87% vs. 81%) and FPG (45% vs. 38%) than females. Females presented a higher percentage of the following MetS components: HDL-C (35% vs. 16%), WC (76% vs. 52%) than males.

Table 3 shows the variables analyzed in subjects with and without MetS, as well as the differences between them. Subjects with MetS had lower MD adherence (35% vs. 54%; *p* = 0.002). Subjects with MetS presented higher baPWV values (16.15 ± 2.58 vs. 15.47 ± 2.49; *p* < 0.001), with no difference in CAVI (*p* = 0.200).

Appendix A (males) and Appendix A (females) in the Appendix A reflect the variables studied in males and females with and without MetS, as well as the differences between them. We found no differences in the percentage of males with MD adherence (*p* = 0.176). Males with MetS presented higher baPWV values (15.96 ± 2.33 vs. 15.55 ± 2.38; *p* = 0.025), with no difference in CAVI (*p* = 0.486). Females with MetS showed lower MD adherence (33% vs. 48%; *p* < 0.001). Females with MetS presented higher baPWV values (16.39 ± 2.63 vs. 15.36 ± 2.64; *p* < 0.001), with no difference in CAVI (*p* = 0.077).

Figure 2 shows the number of MetS components, overall, (a) in males (b) and in females (c).

Figure 3 shows the difference in MD scores of subjects with MetS and subjects without MetS and the different MetS components, overall and by sex.

### 3.2. Association between the Mediterranean Diet and Vascular Stiffness and MetS in People over 65 Years of Age

Table 4 shows the association found between the mean MEDAS screener score with the numbers and components of MetS and the AS values, overall and by sex. In the overall analysis, the mean MD score showed a negative association with the number of MetS components (β = −0.168) and with FPA (β = −0.007); TGC (β = −0.003), WC (β = −0.018), CAVI (β = −0.196), and baPWV (β = −0.065). The mean MD score showed a positive association with the level of HDL-C (β = 0.013). In men, the mean MD score was negatively associated with the number of MetS components (β = −0.181), and with FPA (β = −0.011); TGC (β = −0.004), CAVI (β = −0.230), and baPWV (β = −0.099). In women, the mean MD score was negatively associated with WC (β = −0.026); with the CAVI (β = −0.150) and positively associated with levels of HDL-C (β = 0.017).

Table 5 presents the overall and sex-based logistic regression results. Overall, the adjusted logistic regression models showed that greater MD adherence reduced the likelihood of MetS (OR = 0.675, 95% CI: 0.528 to 0.864) and its components: FPA ≥ 100 mg/dL (OR = 0.640, 95% CI: 0.477 to 0.859); TGC ≥ 150 mg/dL (OR = 0.744, 95% CI: 0.566 to 0.986).

CAVI (OR = 0.835, 95% CI: 0.749 to 0.931), baPWV (OR = 0.939, 95% CI: 0.896 to 0.984), and the increased the likelihood of having HDL-C < 40 mg/dL in males or <50 mg/dL in females (OR = 1.749, 95% CI: 1.305 to 2.326). In males, increased MD adherence reduced the likelihood of TGC ≥ 150 mg/dL (OR = 0.609, 95% CI: 0.417 to 0.891); CAVI (OR = 0.825, 95% CI: 0.711 to 0.957), and baPWV (OR = 0.924, 95% CI: 0.867 to 0.958), and increased the probability of HDL-C < 40 mg/dL (OR = 1.665, 95% CI: 1.082 to 2.563). In females, the adjusted logistic regression models showed that higher MD adherence decreased the likelihood of MetS (OR = 0.605, 95% CI: 0.410 to 0.895) and its components: FPA ≥ 100 mg/dL (OR = 0.604, 95% CI: 0.376 to 0.971) and CAVI (OR = 0.841, 95% CI: 0.713 to 0.991) and increased the likelihood of HDL-C < 50 mg/dL (OR = 1.792, 95% CI: 1.221 to 2.632).

## 4. Discussion

In this study of subjects aged over 65 years, we found that four out of six people adhered to the MD, with a higher MD score in females, and one out of two subjects presented metabolic syndrome, with a higher percentage in females. The association between the MD and vascular stiffness and the nº of components of MetS, as well as its components was negative, except with the blood pressure figures, for which we found no association, and with HDL-cholesterol, for which the association was positive. Finally, the association differed by sex; it was associated with glycemia and triglycerides in men, while in women there was an association of the MD with HDL-C and WC.

In line with previous studies, MD adherence was higher in females than in males [55,56,57,58]. With regard to metabolic syndrome, the figure diagnosed with MetS was 51%, a result which is lower than that published in Hispanics aged over 60 in the USA (57%) [59] and higher than those published in China, 39% [60], or Iran, 37% [61]. The ENRICA study in Spain [57] analyzed 11,149 people, representative of the Spanish population aged over 18, finding a prevalence in those aged 65 or older of 42%. The DARIOS study [58], with 24,670 people between 35 and 74 years of age from ten autonomous communities, found a prevalence of MetS in people aged 65 or over of 47%, with a higher percentage of females with MetS than males (49% males, 54% females, *p* = 0.036). This is in line with other research on people over 65 years of age, such as the ENRICA study [57] (39% males and 44% females) and in the DARIOS study [58] (42% males, 52% females). However, in the USA, no differences were found between males and females in this age group [59]. The frequency of MetS components differed in males and females. Males had a higher percentage of the following MetS components: blood pressure ≥ 130/85 mmHg (87% vs. 81%) and plasma glucose (45% vs. 38%). Females had a higher percentage of HDL-C (35% vs. 16%) and WC (76% vs. 52%) than males, results which are similar to those published by other authors [57,58,62]. However, it should not be forgotten that the prevalence of MetS varies greatly depending on geographical area, age, and sex. Comparing these results, therefore, has important limitations and should be read with caution.

In the overall analysis, this study showed a link between the MD and MetS and its components, except for blood pressure. These results are consistent in the analyses by both multiple and logistic regression. In the elderly Spanish population, a negative relationship was found between MD adherence and the prevalence of MetS [63]. In the PREDIMED-PLUS study [45], which involved 5739 overweight/obese participants aged 55–75 years, it was found that participants with MetS had lower MD adherence. However, in a study of 1404 adults in Luxembourg, an association of the MD with MetS was found when the MD was used as a continuous score, but this association disappeared when the MD was used as a categorical variable [64]. In Hassani et al. [65], MD adherence was not associated with MetS. However, some prospective studies, which investigated the impact of different interventions analyzing the effect of the MD on MetS, did find benefits. This was demonstrated, for example, by a clinical trial with a three-month dietary intervention based on salt restriction and the MD, which resulted in a decrease in the prevalence of MetS compared to the control group [66]. Similarly, in a recently published review, Martemucci et al. [67], showed that MD adherence, together with increased physical activity, were appropriate complementary approaches to preventing the onset of MetS. There are also longitudinal studies which have analyzed the importance of different Mediterranean lifestyle elements measured with MD adherence (MEDLIFE), with a 5-year follow-up of participants in the CORDIOPREV study, finding a lower incidence of MetS and a greater likelihood of reversing MetS, compared to the group with low adherence to MEDLIFE [68]. At 8.7 years of follow-up in the ENRICA study cohort, a Mediterranean lifestyle assessed with the MEDLIFE index was associated with a lower incidence of MetS [44]. However, these studies did not analyze the individual effect of each item, but rather the combined, and possibly synergistic effects of several behaviors related to Mediterranean culture. Therefore, further prospective studies are needed to study the effect that each MD component has on MetS.

In this study, we found an association between the MD and all the components of MetS, apart from blood pressure. Several studies have analyzed the relationship between the MD and the components of MetS, with different results. For example, a logistic regression analysis showed that poor MD adherence was linked to higher WC (OR 1.31) and higher triglyceride levels (OR 2.80) after adjusting for several possible confounding factors [69]. A meta-analysis found that the greater the adherence to the MD, the lower the WC (β = −0.20) and the level of triglycerides (β = −0.27) and the higher the level of HDL-C (β = 0.28), without an association with glycemia or blood pressure [70]. On the other hand, of MD components, higher consumption of vegetables, legumes, and nuts was associated with lower all-cause mortality, while higher vegetable intake was associated with lower cardiovascular mortality, and higher red/processed meat intake was associated with higher cardiovascular mortality in participants with MetS [71]. In sum, high MD adherence can improve the different components making up MetS. However, this may differ from one component to another, probably explained by the heterogeneity of the studies in terms of subjects included, analyses performed, associated pathologies or concomitant treatments. Therefore, further research in this field is necessary.

In the analysis of sex, we found that the MD was associated with glycemia and triglycerides in males, while in females, there was an association of the MD with HDL-C and WC. These associations differ from other studies, such as the one carried out by Hassani et al. [65], which only demonstrated an association of the MD with fasting blood glucose (OR: 0.57) and with abdominal obesity (OR: 0.42) in females. A review of studies on menopausal females concluded that the MD had a beneficial effect on females’ health, leading to a reduction in weight, blood pressure, triglycerides, and total cholesterol [72]. Thus, the association of the MD with different MetS components differs by sex, and, to date, few studies have analyzed this aspect, so more research is needed to clarify the influence of the MD on each of the components by sex.

In the overall analysis, we found that a higher score on the MEDAS questionnaire was associated with lower VS assessed with CAVI and baPWV. Similar results have been found in previous studies [73,74]. Along these lines, the NU-AGE clinical trial [35] found that the MD decreased AS. The trial conducted by the American College of Physicians concluded that the MD could lead to a regression of aortic stiffness [75]. However, the EVasCu study [37] showed that subjects with greater MD adherence had higher VS, although this difference disappeared after adjusting for age. The RoCAV study [76], which analyzed the association between the MD and VS through an assessment of cfPWV in 2640 participants without chronic diseases at the time of recruitment, found no association. Finally, one study found a negative association between the MD and increased CAVI at five years [77]. The discrepancies between the results of the different studies may be understood in the sense that as people age, they tend to have more health problems and adopt healthier dietary patterns accordingly. In addition, arteries lose elasticity with age, increasing AS [29,30]. This may lead to a decrease in the age-related benefits of the MD. Thus, in older people, age may act as a confounding variable in the association between the MD and VS.

In the analysis of sex, the association of the MD with baPWV held only in males. Few studies have analyzed such differences between males and females, and different results were found. Thus, the NU-AGE study [21] did not observe any effect of the intervention on cfPWV. These discrepancies between sexes could be explained by the fact that females have higher MD scores than males [55,56,57,58]. In addition, epigenetic and molecular changes cause an increase in VS and reduce adherence, differing according to sex [29,30,75], all of which are combined with the fact that the differences between sexes in VS are influenced by hormonal and non-hormonal factors [78]. For example, the protection of endogenous estrogens until menopause in females is well known. Meanwhile, in males, VS increases linearly from puberty onwards, indicating that females inherently have stiffer main arteries than males, an effect which is mitigated by sexual steroids during reproductive life. Finally, height, body fat, and other inflammatory factors differ by sex [78]. Further variation is also due to the fact that the stiffness measures used to assess different parts of the vascular tree are also different, with baPWV assessing peripheral AS, while CAVI is a measure of central and peripheral VS [79].

Limitations and strengths: This study has important limitations: 1. It is a cross-sectional study, and causality cannot therefore be established; 2. The data analyzed comes from three research studies, with different participant characteristics; 3. The origin of the participants from the urban population prevents generalization of results to the rural population; 4. The use of a cut-off point for MD adherence is arbitrary; 5. The data on MD consumption was recorded using questionnaires. Nevertheless, the study also has some strengths, which include: the large sample size, the use of a standardized protocol for carrying out measurements, the fact that the researchers who performed them had prior training, and that measurements were made with validated and calibrated devices.

## 5. Conclusions

The results yielded by this study indicate that greater adherence to the Mediterranean diet decreases VS and the percentage of subjects with MetS, although results differ in the associations found with MetS components according to sex.

## Figures and Tables

**Figure 1 nutrients-16-03464-f001:**
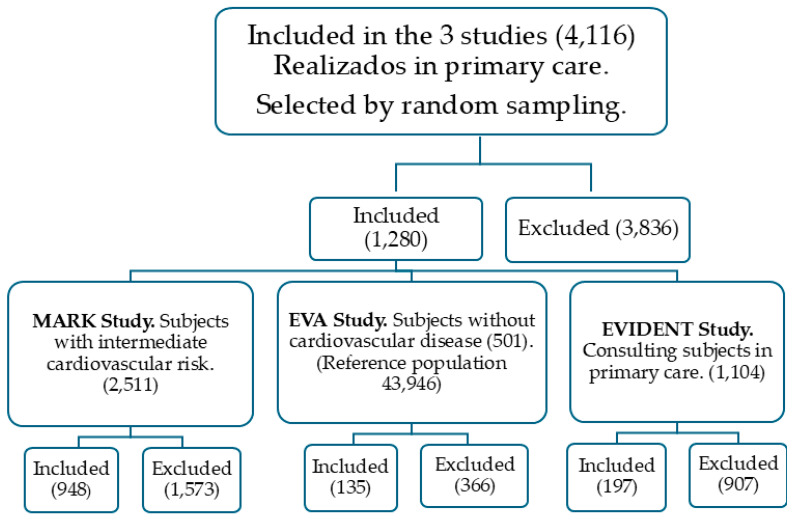
Number of subjects included and excluded from each of the three studies.

**Figure 2 nutrients-16-03464-f002:**
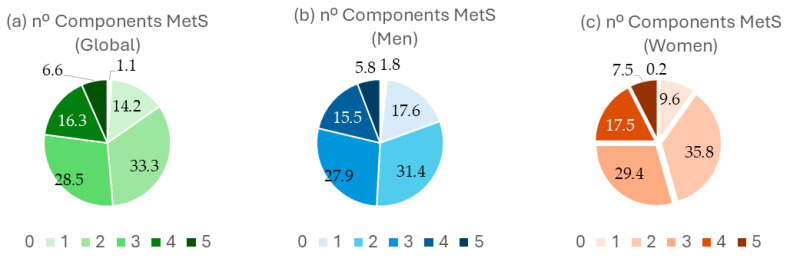
% of participants according to the nº of MetS components, overall and by sex. nº: number of subjects; MetS: Metabolic syndrome; MD: Mediterranean diet.

**Figure 3 nutrients-16-03464-f003:**
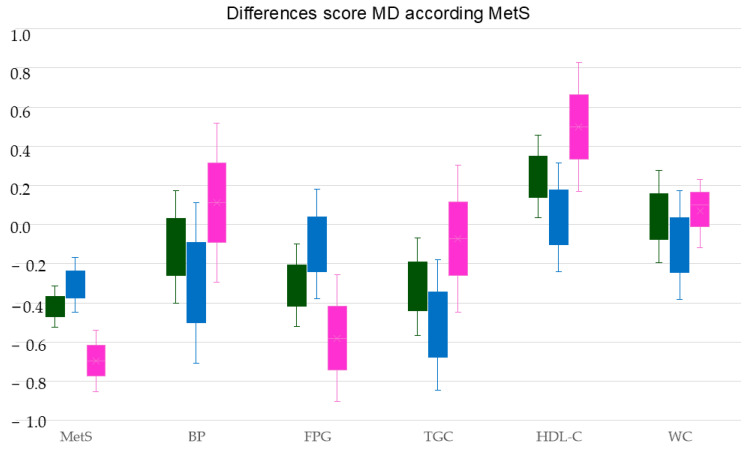
Difference in MD scores between subjects with and without MetS and MetS components, overall and by sex. MetS: Metabolic syndrome; MD: Mediterranean diet; BP: blood pressure; FPG: fasting plasma glucose; TGC: triglycerides; HDH-C: high-density lipoprotein cholesterol; WC: waist circumference. Green bars represent global differences, blue bars represent differences in men, and pink bars represent differences in women.

**Table 1 nutrients-16-03464-t001:** Diagnostic criteria for MetS.

Blood pressure	BP figures ≥ 130/85 mmHg or consumption of antihypertensives
Glycemia	FBG ≥ 100 mg/dL or or consumption of hypoglycemic medication
Triglycerides	TGC ≥ 150 mg/dL or lipid-lowering medication
HDL-cholesterol	HDL-C < 40 mg/dL in males or <50 mg/dL in females
Waist circumference	WC ≥ 88 cm in females or WC ≥ 102 cm in males

MetS: Metabolic syndrome, BP: blood pressure; FBG: fasting plasma glucose, TGC: triglycerides, HDL-C: high–density lipoprotein cholesterol, WC: waist circumference. Adapted from International Diabetes Federation Task Force on Epidemiology and Prevention [37].

**Table 2 nutrients-16-03464-t002:** Characteristics of the study participants.

	Overall (nº = 1280)		Males (nº = 736)	Females (nº = 544)		*p* Value	
Mediterranean Diet	Mean or nº	SD or (%)	Mean or nº	SD or (%)	Mean or nº	SD or (%)	
MD (total score)	6.00	1.90	5.92	1.92	6.11	1.88	0.036
MD adherence, n (%)	504	(39)	288	(40)	216	(40)	0.440
Risk factors							
Age, (years)	69.52	3.58	69.47	3.51	69.58	3.67	0.286
SBP, (mmHg)	136.81	17.95	138.86	17.42	134.03	18.31	<0.001
DBP, (mmHg)	80.15	10.06	80.96	9.67	79.04	10.48	<0.001
Hypertension, n (%)	947	(74)	548	(74%)	399	(73%)	0.350
Antihypertensive drugs, n (%)	739	(58)	410	(56)	329	(61)	0.049
HDL cholesterol, (mg/dL)	54.10	14.79	50.96	12.79	58.33	16.20	<0.001
Triglycerides, (mg/dL)	121.43	57.56	120.87	60.49	122.18	53.38	0.344
Dyslipidemia, n (%)	1042	(81)	564	(77)	478	(88)	<0.001
Lipid–lowering drugs, n (%)	463	(36)	240	(33)	223	(41)	0.001
FPG, (mg/dL)	103.04	28.91	104.37	28.61	101.23	29.23	0.028
Diabetes mellitus, n (%)	298	(23)	180	(25)	118	(22)	0.142
Hypoglycemic drugs, n (%)	261	(20)	155	(21)	106	(19)	0.268
WC, cm	99.61	10.94	102.59	9.42	95.57	11.55	<0.001
Obesity, n (%)	391	(30)	206	(28)	185	(34)	0.012
Vascular stiffness							
CAVI	9.30	1.11	9.49	1.05	9.03	1.13	<0.001
baPWV, m/s	15.82	2.56	15.75	2.46	15.92	2.68	<0.001
MetS and its components							
Number of MetS components	2.64	1.14	2.55	1.17	2.77	1.09	0.129
Mets, n (%)	658	(51)	362	(49)	296	(54)	0.036
BP ≥ 130/85 mmHg, n (%)	1082	(84)	639	(87)	443	(81)	0.005
FPG ≥ 100 mg/dL, n (%)	541	(42)	334	(45)	207	(38)	0.005
TGC ≥ 150 mg/dL, n (%)	289	(23)	162	(22)	127	(23)	0.303
HDL-C mg/dL < 40 males, <50 females, n (%)	308	(24)	119	(16)	189	(35)	<0.001
WC ≥ 88 cm females, ≥102 cm males, n (%)	798	(62)	385	(52)	413	(76)	<0.001

Values are the mean and standard deviations for continuous data, and numbers and proportions for categorical data. n: number of subjects; MetS: Metabolic syndrome; MD: Mediterranean diet; SBP: systolic blood pressure; DBP: diastolic blood pressure; HDL-C: high-density lipoprotein cholesterol; FPG: fasting plasma glucose; WC: waist circumference; BP: blood pressure; TGC: triglycerides; CAVI: cardio-ankle vascular index; baPWV: brachial-ankle pulse wave velocity; *p* value: differences between males and females.

**Table 3 nutrients-16-03464-t003:** Characteristics of the subjects included with and without MetS.

	With MetS (nº = 658)	Without MetS (nº = 622)		*p* Value	
Mediterranean Diet	Mean or N	SD or (%)	Mean or N	SD or (%)	
MD (total score)	5.85	1.81	6.16	1.98	0.003
MD adherence, n (%)	232	(35.3)	272	(54.0)	0.002
Risk factors					
Males, n (%)	362	(51)	374	(49)	0.236
Females, n (%)	296	(54)	248	(46)	0.036
Age, (years)	69.46	3.31	69.58	3.84	0.534
SBP, (mmHg)	139.08	16.38	134.41	19.21	<0.001
DBP, (mmHg)	81.16	9.85	79.07	10.18	<0.001
Hypertension, n (%)	562	(85)	385	(62)	<0.001
Antihypertensive drugs, n (%)	447	(68)	292	(47)	<0.001
HDL cholesterol, (mg/dL)	47.78	11.23	60.79	15.17	<0.001
Triglycerides, (mg/dL)	146.26	64.78	95.12	32.17	<0.001
Dyslipidemia, n (%)	534	(82)	508	(81)	0.411
Lipid–lowering drugs, n (%)	268	(41)	195	(31)	<0.001
FPG, (mg/dL)	114.28	32.28	91.16	18.55	<0.001
Diabetes mellitus, n (%)	248	(38)	50	(8)	<0.001
Hypoglycemic drugs, n (%)	222	(34)	39	(6)	<0.001
WC, cm	103.37	10.83	95.64	9.56	<0.001
Obesity, n (%)	291	(44)	100	(26)	<0.001
Vascular stiffness					
CAVI	9.33	1.12	9.25	1.10	0.200
baPWV, m/s	16.15	2.58	15.47	2.49	<0.001
MetS and its components					
Number of MetS components	3.57	0.71	1.66	0.52	<0.001
BP ≥ 130/85 mmHg, n (%)	621	(94)	461	(74)	<0.001
FPG ≥ 100 mg/dL, n (%)	445	(68)	95	(15)	<0.001
TGC ≥ 150 mg/dL, n (%)	270	(41)	19	(7)	<0.001
HDL-C mg/dL < 40 males, < 50 females, n (%)	268	(40)	40	(13)	<0.001
WC ≥ 88 cm females, ≥102 cm males, n (%)	517	(78)	281	(35)	<0.001

Values are the mean and standard deviations for continuous data, and the numbers and proportions for categorical data. n: number of subjects; MetS: metabolic syndrome; MD: Mediterranean diet; SBP: systolic blood pressure; DBP: diastolic blood pressure; HDL-C: high-density lipoprotein cholesterol; FPG: fasting plasma glucose; WC: waist circumference; BP: blood pressure; TGC: triglycerides; CAVI: cardio-ankle vascular index; baPWV: brachial-ankle pulse wave velocity; *p* value: differences between subjects with and without MetS.

**Table 4 nutrients-16-03464-t004:** Association of the MD with vascular stiffness and with the nº and components of the metabolic syndrome, overall and by sex.

Global	β	(95%	CI)	R^2^	*p*
Number of MetS components	−0.168	(−0.269	to −0.068)	3.30	0.001
SBP, (mmHg)	0.001	(−0.005	to 0.007)	2.90	0.669
DBP, (mmHg)	−0.005	(−0.015	to 0.006)	3.00	0.402
FPG, (mg/dL)	−0.007	(−0.012	to −0.003)	3.70	0.001
Triglycerides, (mg/dL)	−0.003	(−0.005	to −0.002)	3.90	<0.001
HDL cholesterol, (mg/dL)	0.013	(0.006	to 0.020)	3.90	<0.001
WC, cm	−0.018	(−0.028	to −0.008)	3.90	<0.001
CAVI	−0.196	(−0.294	to −0.099)	4.00	<0.001
baPWV, m/s	−0.065	(−0.107	to −0.060)	3.30	0.002
Females					
Number of MetS components	−0.115	(−0.282	to 0.051)	5.20	0.174
SBP, (mmHg)	0.004	(−0.004	to 0.013)	5.00	0.329
DBP, (mmHg)	−0.012	(−0.027	to 0.002)	4.90	0.099
FPG, (mg/dL)	−0.001	(−0.009	to 0.006)	4.00	0.681
Triglycerides, (mg/dL)	−0.002	(−0.005	to 0.001)	5.00	0.302
HDL cholesterol, (mg/dL)	0.017	(0.008	to 0.027)	7.10	0.001
WC, cm	−0.026	(−0.040	to −0.012)	6.40	<0.001
CAVI	−0.150	(−0.293	to −0.008)	5.50	0.038
baPWV, m/s	−0.021	(−0.082	to 0.040)	4.00	0.499
Males					
Number of MetS components	−0.181	−0.307	to −0.056	4.50	0.005
SBP, (mmHg)	−0.002	−0.009	to 0.006	3.50	0.702
DBP, (mmHg)	0.003	−0.012	to 0.018	3.80	0.663
FPG, (mg/dL)	−0.011	−0.016	to −0.005	4.50	<0.001
Triglycerides, (mg/dL)	−0.004	−0.006	to −0.002	5.10	<0.001
HDL cholesterol, (mg/dL)	0.006	−0.005	to 0.017	3.70	0.281
WC, cm	−0.009	−0.024	to 0.006	3.00	0.230
CAVI	−0.230	−0.363	to −0.097	4.80	0.001
baPWV, m/s	−0.099	−0.155	to −0.043	4.30	0.001

Multiple regression analysis using as dependent variables: vascular stiffness, number of components, SBP, DBP, FPG, triglycerides, HDL cholesterol and WC; Mediterranean diet score was the independent variable; adjustment variables: age, sex, and use of antihypertensive drugs, hypoglycemic and lipid-lowering agents. MetS: metabolic syndrome; MD: Mediterranean diet; SBP: systolic blood pressure; DBP: diastolic blood pressure; HDL: high-density lipoprotein; FPG: fasting plasma glucose; WC: waist circumference; CAVI: cardio-ankle vascular index; baPWV: brachial-ankle pulse wave velocity. β: non-standardized coefficient with its confidence intervals. R^2^: Represents the for variance accounted for by predictors in percentage.

**Table 5 nutrients-16-03464-t005:** Association of Mediterranean diet adherence with vascular stiffness and metabolic syndrome and its components, overall and by sex; Logistic regression analysis.

Global.	OR	(95%	CI)	R^2^	*p*
MetS	0.675	(0.528	to 0.864)	2.50	0.002
BP ≥ 130/85 mmHg	0.932	(0.651	to 1.335)	1.50	0.701
FPG ≥ 100 mg/dL	0.640	(0.477	to 0.859)	2.40	0.003
Triglycerides ≥ 150 mg/dL	0.747	(0.566	to 0.986)	1.90	0.040
HDL-C mg/dL < 40 males, <50 mg/dL females	1.749	(1.315	to 2.326)	3.10	<0.001
WC ≥ 88 cm females, ≥102 cm males	0.815	(0.645	to 1.553)	1.50	0.088
CAVI	0.835	(0.749	to 0.931)	2.50	0.001
baPWV, m/s	0.939	(0.896	to 0.984)	2.10	0.008
Females					
MetS	0.605	(0.410	to 0.895)	6.00	0.012
BP ≥ 130/85 mmHg	1.347	(0.770	to 2.356)	4.70	0.296
FPG ≥ 100 mg/dL	0.604	(0.376	to 0.971)	5.50	0.033
Triglycerides ≥ 150 mg/dL	1.035	(0.681	to 1.573)	4.50	0.873
HDL-C mg/dL < 40 males, <50 females	1.792	(1.221	to 2.632)	6.70	0.003
WC ≥ 88 cm females, ≥102 cm males	0.751	(0.495	to 1.140)	4.50	0.178
CAVI	0.841	(0.713	to 0.991)	5.30	0.038
baPWV, m/s	0.952	(0.887	to 1.022)	4.70	0.171
Males					
MetS	0.760	(0.552	to 1.046)	3.20	0.093
BP ≥ 130/85 mmHg	0.685	(0.423	to 1.109)	3.10	0.124
FPG ≥ 100 mg/dL	0.696	(0.477	to 1.015)	3.30	0.060
Triglycerides ≥ 150 mg/dL	0.609	(0.417	to 0.891)	3.90	0.011
HDL-C mg/dL < 40 males, <50 females	1.665	(1.082	to 2.563)	3.70	0.020
WC ≥ 88 cm females, ≥102 cm males	0.840	(0.622	to 1.134)	2.90	0.255
CAVI	0.825	(0.711	to 0.957)	3.70	0.011
baPWV, m/s	0.924	(0.867	to 0.985)	3.60	0.015

Logistic regression analysis using as dependent variables MetS and its components, as independent variable Mediterranean diet score ≥ 7, and as adjustment variables age, sex, and use of antihypertensive drugs, hypoglycemic and lipid-lowering agents. MD, Mediterranean diet; MetS, metabolic syndrome; BP, blood pressure ≥ 130/85 mmHg; FPG, fasting plasma glucose ≥ 100 mg/dL, TGC, triglycerides ≥ 150 mg/dL; HDL-C, high–density lipoprotein < 40 males mg/dL, <50 mg/dL females; WC, waist circumference WC ≥ 88 cm females, ≥102 cm males.

## Data Availability

The data supporting the findings of this study are available on ZENODO under https://zenodo.org/records/12166167.

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
