# Peer review of "The Relationship between the Mediterranean Diet and Vascular Stiffness, Metabolic Syndrome, and Its Components in People over 65 Years of Age"

_nutrients, 2024, doi:10.3390/nu16203464_

Round 1

Reviewer 1 Report

Comments and Suggestions for Authors

Dear Corresponding Author, thank you for submiting your paper and congratulations on your work.

Brief summary:
Your study examines the relationship between adherence to the Mediterranean diet (MD), arterial stiffness and metabolic syndrome (MS) in a population of over 1,200 individuals aged over 65 years. The main objective is to analyze these associations both overall and by gender. The main results show that greater adherence to MD is associated with lower arterial stiffness and a lower probability of MS, with some interesting differences between men and women. The study significantly contributes to understanding the role of the Mediterranean diet in cardiovascular health of the elderly.

General comments: The manuscript is well structured and presents a solid metodology. The use of data from three previous studies (EVA, MARK and EVIDENT) increases the robustness of the sample. The statistical analysis is appropriate and well executed, using multiple linear regression and logistic regression models that allow for control of potential confounding factors.

The results are presented clearly and in detail, with tables and figures that effectively support the text. The discussion is well articulated and places the results in the context of existing literature.

However, there are some points that could be improved:

  • The introduction, although complete, could benefit from a clearer definition of the gaps in current literature that this study intends to fill, especially from line 95.
  • In the discussion, it would be useful to explore more deeply the possible reasons for the observed differences between men and women, especially in reference to lines 307 and 312.

Specific comments:

  • Table 2: Some p values are reported as "<0.001", while others as "0.001". For consistency, it is advisable to use the same format throughout the table.
  • Figure 2: The labels on the y-axis are difficult to read. Consider increasing the font size or rotating the graph. The images are small and the numbers inside are not perfectly readable, I believe that the graphics are also necessary for an excellent understanding of the paper.
  • Lines 260-262: The sentence "the mean MD score was negatively associated with the number of MetS components (β = -0.181, 95% CI: -0.307 to -0.056) and with the following MetS components: FPA (β = -0.011; 95% CI: -0.016 to -0.005), TGC (β = -0.004; 95% CI: -0.006 to -0.002), and with AS values: CAVI (β = -0.230; 95% CI: -0.363 to -0.097) and baPWV (β = -0.099; 95% CI: -0.155 to -0.043)." is really very long and could be divided to improve readability, now it is really difficult.

In conclusion, your study provides a valid and interesting contribution to the literature on Mediterranean diet and cardiovascular health in the elderly. With some minor revisions, it will be a valuable addition to research in this field and certainly of interest to the journal Nutrients.

I look forward to reading a definitive work.

Reviewer 2 Report

Comments and Suggestions for Authors

1. In “statistical analysis” section, the use of “adjustment variables: age, sex, and use of antihypertensive drugs, hypoglycemic and lipid-lowering agents” should be explained why these variables were considered as adjustment variables.

2. In Figure 1, the sum of the subjects included was 948+135+195=1278, not equal to 1280 as reported.

3. In Figure 2, the six components of MetS should be denoted below the figure, such as 0 = BP ≥ 130/85 mmHg, 1= , 2=…,5= to help understand the figure.

4. In Figure 3, it was a little difficult to understand. What is the vertical axis mean? How can BP be calculated from SBP and DBP? What did the three colors mean? Please denote them in the Figure or someplace.

5. In Table 4, what does the R2 mean? If it stands for variance accounted for by predictors, then it should never be beyond 1.0. Did the author present the values with %? If so, please denote it. Besides, the regression coefficient “β” stands for the unstandardized or standardized ones? And its CI for which one?

6. In Table 5, for logistic regression, it should not have the R2 in theory. Please check it.

7. In some Tables or Figures, there was the signal “n0”, what does it mean?

Round 2

Reviewer 2 Report

Comments and Suggestions for Authors

OK